# Evaluating Organizational Guidelines for Enhancing Psychological Well-Being, Safety, and Performance in Technology Integration

Federico Fraboni , Hannah Brendel and Luca Pietrantoni *

Department of Psychology, Alma Mater Studiorum, University of Bologna, 40126 Bologna, Italy;
federico.fraboni3@unibo.it (F.F.); hannah.brendel3@unibo.it (H.B.)
* Correspondence: luca.pietrantoni@unibo.it

**Abstract:** Organizations that integrate new technologies, such as collaborative robots, often struggle to maintain workers' psychological well-being during transitions. Integrating new technologies can, in fact, negatively impact job satisfaction, motivation, and organizational culture. It is thus essential to prioritize workers' psychological sustainability to benefit fully from these technologies' advantages, such as reduced production times and increased flexibility. This study evaluates the impact of eight guidelines designed to support organizations in optimizing human–robot collaboration. The guidelines focus on safety, training, communication, worker agency, and stakeholder involvement. We investigated possible implementation solutions and assessment methods or KPIs for each guideline. We conducted an online survey targeting experts in robotics to gather opinions on the guidelines' potential impact on workers' psychological well-being, safety, and performance. The survey also asked about implementation solutions and KPIs for evaluating their effectiveness. Proposed solutions, such as demonstration videos and hands-on training, have the potential to enhance users' perceived safety and confidence in the system. KPIs, such as subjective perceived safety, risk assessment, and user satisfaction, can be employed to assess the success of these implementations. The study highlights key strategies for ensuring workers' psychological well-being, optimizing performance, and promoting a smooth integration of robotic technologies. By addressing these factors, organizations can better navigate technology integration challenges, fostering a more sustainable and human-centric approach to deploying robotic systems in the workplace.

**Keywords:** human–robot collaboration; psychological sustainability; psychological well-being; organizational guidelines; technology integration; Industry 4.0

## 1. Introduction

Organizations are experiencing rapid changes in the way they operate, with the integration of new technologies, such as collaborative robotic systems, becoming increasingly prevalent [1]. While this can bring many benefits, such as reduced production times and increased flexibility, it can also bring challenges. As organizations incorporate new technologies, it is essential to prioritize the psychological sustainability of workers during the transition. Ensuring workers are equipped with the necessary skills and knowledge to interact effectively with new technologies is crucial for their psychological well-being and the organization's success. Workers' psychological well-being refers to the positive mental state of employees concerning their work and workplace. It encompasses various dimensions: job satisfaction, motivation, work–life balance, and psychological safety. A high level of psychological well-being among employees is associated with better job performance, increased job satisfaction, and reduced stress and burnout [2]. Thus, it is essential to acknowledge that integrating new technologies is both a technical challenge and a social and psychological one, impacting workers' job satisfaction, motivation, and organizational

culture. Therefore, it is crucial for organizations to ensure that workers remain motivated and engaged and that the organization can realize the full benefits of the new technology.

Advancements in industrial and digital technology within Industry 4.0 have led to increased complexity in both technical and organizational systems, resulting in new levels of socio-technical interaction [3,4]. Industry 4.0 is a term used to describe the fourth industrial revolution characterized by integrating advanced technologies such as the Internet of Things (IoT), artificial intelligence (AI), robotics, and data analytics into the manufacturing industry. This technological shift is expected to revolutionize how products are designed, produced, and delivered, leading to greater efficiency, productivity, and customization. Industry 4.0 also involves digitizing the entire value chain, from suppliers to customers, improving collaboration, transparency, and responsiveness. Overall, Industry 4.0 represents a significant shift in manufacturing companies' operations, focusing on leveraging technology to drive innovation, growth, and competitiveness [5]. These transformations significantly benefit organizations, including reduced production times, exploring new business models, and making production more hybrid, flexible, and autonomous [6].

Organizations utilize technology to enhance manufacturing firms' productivity, efficiency, and working conditions. One major technology implemented in the context of Industry 4.0 is collaborative robotics [7]. A collaborative robot (cobot) is designed to work with humans in a shared workspace, performing tasks requiring human and robotic skills. Cobots are, in fact, considered to be complementary to human workers, utilizing their unique strengths, including versatility and analytical capabilities [8–10]. Instead of replacing workers, as traditional industrial robots are meant for, cobots are designed to increase workers' skills, maintain proficiency due to aging or disability, and increase equality by aiding people previously unsuitable for specific tasks. Unlike traditional industrial robots, which are often kept in cages to separate them from human workers, cobots are safe and flexible enough to work in close proximity to humans without causing harm. Cobots are used in various industries, including manufacturing, healthcare and logistics, to perform tasks such as assembly, packaging, inspection, and material handling. They can also assist workers with physically demanding or repetitive tasks. Examples of humans and cobots working together include: (a) in manufacturing, a human worker can assemble a product while a cobot holds the parts in place and performs some of the assembly steps; (b) in a hospital, a cobot might assist a surgeon during a procedure by handing him tools or holding tissue in place; (c) in a warehouse, a cobot could work with human workers to pick and pack items for shipment; (d) in a research lab, a cobot could work with scientists to perform experiments, such as handling samples or taking measurements. Overall, cobots can potentially increase productivity, safety, and job satisfaction by enabling humans and robots to work together more effectively.

However, the ongoing automation and digital transformation in the light of Industry 4.0 have resulted in a continuously increasing number of cobots in organizations, resulting in significant changes in work characteristics, including tasks, work environment, teamwork, and work organization. As a result, introducing new technology within an organization can potentially change the socio-organizational system [11].

## 1.1. Positive and Negative Psychological Effects of Collaboration with Robots

Generally, prior research has shown both positive and negative effects of collaboration with robots on job quality. On the one hand, research has demonstrated that collaboration can improve employees' work environments if cobots take over highly repetitive or high-risk elements of a specific task. Such task-sharing between human workers and robots may significantly reduce physical demands and task load, improving employees' job satisfaction and psychological well-being [12].

However, collaboration with robots may result in a perceived loss of control due to a shift in task allocation and increased dependency on the robot, negatively affecting perceived autonomy and increasing stress levels, adversely impacting workers' cognitive workload. This implies that adopting collaborative robotics may significantly impact

workers' safety, level of satisfaction, cognitive demand, motivation [13], and anxiety [14]. Consequently, it has become more critical for organizations to consider robots' impact on human resources, as even minor changes in the work environment and tasks may significantly affect an individual's perceived work experience and psychological well-being [15].

Prior research has shown that the effect on safety and psychological well-being is contingent upon different design variables, including workstation layout and elements, robot system features, performance, and organizational measures. Most of the literature in the context of Industry 4.0 focuses on the technological perspective, overlooking psychological aspects that accompany the introduction of Industry 4.0 in organizations and its effects on work systems, organizations' and workers' preparation, and psychological well-being [16]. A technological change within an organization simultaneously comes with a shift in an organization's practices and may result in organizational challenges, ultimately affecting human workers [17–19].

For managers, a successful deployment entails making the most of the combined strengths of both human and robotic resources [20]. This allows achieving impact on three primary indicators: (a) organizational performance, understood as increased productivity and quality of products or services; (b) safety, in terms of reduced errors and the number of safety adverse events; and (c) workers' psychological well-being, encompassing job satisfaction, increased motivation, and engagement.

### 1.2. Organizational Measures to Manage the Integration of Robotic Systems

Over the past years, researchers have explored the role and impact of organizational factors and measures in the context of introducing cobots across various organizations and sectors. Charalambous et al. [21–23] synthesized evidence regarding individual and organizational factors and developed a roadmap for successfully introducing cobots in industries. The roadmap comprises two main propositions: a training program highlighting the robot's key characteristics, such as perceived safety and reliability, to help operators build trust with their robot teammates, and operator empowerment, which is essential as operators gain experience working with the robot, especially during events like robot failures, errors, or deviations.

Moreover, Charalambous et al. [22] devised a theoretical framework encompassing the critical organizational factors relevant to the new technology adoption. They conducted an exploratory case study to determine whether these factors could serve as enablers or barriers. The authors pinpointed seven key organizational factors to consider for the implementation of industrial HRC when collaborative technologies were still emerging: (1) communication of the change to employees, (2) operator participation in implementation, (3) training and development of the workforce, (4) the presence of a process champion, (5) organizational flexibility through employee empowerment, (6) senior management commitment and support, and (7) the impact of union involvement. Berx et al. [24] examine organizational factors as risk factors, such as the broader acceptance of robots by management within the organization and, more broadly, organizational structures, policies, and processes that may pose a risk (e.g., lack of training, deskilling, work design and working times, and insufficient communication of the technology agenda).

In the upcoming sections, the present contribution focuses on identifying and describing the variables identified in the literature as essential factors for implementing cobots in the workplace. Specifically, these factors have been found to promote worker well-being, safety, and performance in human–robot collaboration. To combine these variables, we propose the framework of the systems approach, which considers the different components of a system and how they interact with each other [25]. It is a framework for understanding how different system components interact and how changes in one component can affect the system as a whole. The systems approach has been applied to various fields, including engineering, ecology, and healthcare. In the case of this study, the system would be the

implementation of cobots in the workplace, and the different variables would be considered subsystems that interact with each other.

### 1.3. Allocation of Task and Job Motivation and Satisfaction

The work and organizational psychology literature has long demonstrated that effectively managing each team member's unique skills contributes to a team's success [26]. Additionally, the importance of creating a person–job fit by aligning an individual's qualifications with job demands has been emphasized, as it positively influences job satisfaction and reduces the intention to quit [27]. Therefore, when implementing HRC within an industry, it is essential to identify the appropriate allocation of tasks between humans and robots, determining where robots may completely replace human work and where they should merely complement it [28].

In general, cobots can perform monotonous, repetitive, and dangerous tasks, reducing the risk of workplace injuries and accidents. Human workers remain necessary due to their advanced cognitive skills, allowing them to handle tasks that require high reaction time, manage unpredictable situations, or deal with complex assembly settings [27]. However, it is crucial to consider that introducing cobots may reduce workers' autonomy and skills, potentially leading to increased stress, counterproductive work behaviors, and demotivation [1]. As a result, it is vital to design a system that enhances the value of workers' activities without eliminating their expertise [11].

Pauliková et al. [29] conducted a SWOT analysis on the impacts of increased robotization on jobs in industrial organizations. They identified a lack of qualified workers as a significant threat and an appropriate combination of human and robot skills in collaborative work as a primary opportunity. The authors also emphasize the importance of establishing the right balance between robots' strengths and human skills. However, current scenarios often focus on technical aspects for task allocation, primarily considering the robot's capabilities rather than the human's, for example, leaving human workers with leftover tasks. In contrast, Ranz et al. [28] propose a task allocation approach that combines the above-mentioned criteria while improving work quality and job satisfaction by considering both robots' and workers' actual capabilities. The task allocation process needs to be capacity-based, focusing on the individual worker rather than the entire workforce and allocating tasks according to individual competence levels with an approach allowing ad hoc reorganization to mitigate the risk of deskilling [30].

Considering the above aspects, there is no single solution for human–robot task allocation. However, recurring themes across different task allocation approaches emphasize first decomposing the work process into tasks and then performing human–robot task allocation. By following a suitable human–robot task allocation approach, organizations can effectively leverage the skills of both workers and robots to improve work quality, job satisfaction, and safety.

### 1.4. Employees Participation

Employee participation in decision-making and implementation processes within organizations has been shown to encourage supportive behavior [31,32]. When employees are involved in change processes, they may develop a sense of control and ownership for the impending change, increasing their readiness for it [33,34].

Insights from organizational change literature on the significance of employee participation can be applied to the context of the implementation of robotic systems. Employee participation can lead to greater acceptance of the new system, enabling thorough analysis of work activities due to workers' unique knowledge of tasks and processes. Charalambous et al. [22] demonstrated in their case study that operator participation facilitated the change process, contributing to the successful development and implementation of automation. Interviewees suggested that earlier involvement, starting from the conceptual design stage, could have led to an even smoother implementation and prevented problems.

Several approaches, such as user-centered design, emphasize end-users' needs throughout the entire design process. Although not specifically tailored for designing collaborative human–robot systems, these approaches share common features, such as involving end-users from the start, evaluating progress based on real needs, and incorporating feedback on newly implemented technologies [35]. For example, the International Organisation for Standardisation (ISO 9241-210) [36] advises practitioners to follow human-centered design principles and actions to enhance human-system interaction when planning and managing the design process for computer-based interactive systems. These principles include understanding current tasks, stakeholders, and the environment, involving users throughout the design and development process, evaluating design using user feedback, utilizing an iterative design process, and incorporating a multidisciplinary team.

In a study by Colim et al. [20], a participatory lean approach was employed to improve ergonomics for assembly workers in a collaborative robotic workstation. The study considered human factors, performance indicators, ergonomics assessment, workers' perceptions of using robots in industry, ergonomic improvements, well-being, and acceptability of the new preassembly workstation. Results indicated that workers were satisfied and motivated with the newly implemented collaborative workstation design. The authors emphasized the importance of worker involvement in the design and implementation process for identifying and addressing potential issues.

Employee participation is crucial for organizations to consider when introducing technological changes, as it can enhance employee health, well-being, and system efficiency [37]. Furthermore, scholars advocate for a shift from a technology-oriented to a human-oriented design approach, where system-wide aspects extend beyond technological considerations. In this process, employees can serve as subject matter experts, complementing the design team's knowledge with their unique, field-specific expertise [38].

### 1.5. Training and Development in the Organizations

When organizations implement new technologies that affect workers' tasks, responsibilities, and demands, it is crucial to provide adequate support and identify transversal and professional skills that enable workers to effectively cope with the newly introduced technologies, such as cobots [39,40]. The transition to more knowledge-intensive work can be challenging for the workforce. Organizations need to invest in personnel training, lifelong learning, and ongoing development programs to help workers adapt to new work demands and digital transformation, which can enhance workers' awareness, skills, commitment, and safety while reducing stress and turnover [41].

Workforce training is a crucial enabler for the successful implementation of HRC in the industry. Providing training to selected workers can increase their ownership and confidence in working with the system and promote knowledge sharing among fellow workers, leading to higher acceptance of the new automation. However, no one-size-fits-all training system exists, and organizations can use various training techniques depending on the context and employee skill level [42]. On-the-job training, including simulation-based training and training on the equipment, can be effective but cost-intensive and time-consuming [43]. Video-based training, such as instructional videos, computer-based simulations, or virtual reality-based training, can provide flexibility and less risk. Classroom-based training can include lectures, demonstrations, and hands-on exercises to provide a general understanding of the robot's capabilities, limitations, and safe operating procedures. Self-paced online training, such as online tutorials and interactive training modules, offers adaptive learning paths for experienced and inexperienced workers alike [44]. Collaborative training techniques, where employees work with robots and learn from each other through trial and error, can also be effective [45].

Training is an essential organizational measure for addressing skill gaps when introducing new systems. However, a sustainable learning and development solution is necessary for long-term success, emphasizing lifelong learning and worker development

initiatives. The most suitable training technique will depend on the specific context, industry, task, and employee skill level.

*1.6. Empowerment and Knowledge Sharing*

Employee empowerment is crucial for successfully implementing cobots within an organization. Integrating complex hardware and software may result in unforeseen challenges, leaving workers uncertain [37]. Consequently, the literature suggests transitioning from a conventional management hierarchy to a structure characterized by a shared knowledge stream, where decision-making is delegated to lower levels, such as robot operators, leading to a more blurred management hierarchy.

Adjusting the control structure allows robot operators to better understand task requirements and the robotic system, especially during incidents like robot failures or deviations. Empowering workers in such situations enables them to update and refine their mental model of the robot based on prior collaboration experience. However, employee empowerment needs to be considered within a specific context, with experts still available for support when necessary [22,23].

To avoid developing a blame culture when system malfunctions occur, it is essential to establish a learning culture that allows employees to make mistakes. This approach mitigates the risk of workers becoming passive when troubleshooting robotic system issues, instead of waiting for experts to resolve them. Moreover, incorporating workers' knowledge into training initiatives for new operators accelerates trust calibration between humans and robots [23]. In conclusion, fostering a flexible organizational structure and a shared knowledge stream empowers employees during the introduction of a new system.

*1.7. Management Support*

Management support can play a crucial role in helping employees understand the benefits of change and shape their perceptions, compliance, and readiness for change, ultimately resulting in either support or resistance behavior. Senior management support is particularly important when introducing technological change within an organization [46,47].

Senior management support and commitment have been identified as significant enablers that positively influence workers' perceptions of a project's credibility and significance [47,48]. Robotic operators experiencing senior management support may feel more acknowledged for their efforts, potentially increasing their morale. This support can be particularly important during implementation when confronted with obstacles, as senior management may act as a protective factor against project stagnation.

There are several ways in which senior management can show support to their employees during new system implementation. One practical approach is to allocate necessary resources for developing an automated system, such as providing financial resources, personnel, or employee training and education. Clear communication of decisions concerning resource allocation can increase stakeholders' acceptance. Lai et al. [49] identified critical success factors when implementing advanced automation and robotics processes within the manufacturing industry, suggesting further actions for senior management to show support. These actions include having a good understanding and knowledge of the introduced robotic system and automation implementation and providing leadership to workers (either directly or through delegation of authority) to ensure smooth project implementation and organizational members' cooperation. Senior management should also establish directives and policies needed for successful project implementation and link the robotics and automation implementation with the organization's strategy and guide strategic decision-making [50].

Organizational communication has been identified as a critical factor for the success of change actions, as a formal communication strategy may enhance organizational members' commitment and supportive behavior towards the change while reducing their uncertainty [48,51]. Insufficient information can result in reduced well-being, anxiety, and unfounded concerns. Continuous communication, such as regular updates on the

change process, can help understand employees' perceptions and emotions, reduce adverse reactions, and provide timely information [52].

In summary, management support is critical when integrating a novel technology within an organization. Senior management can boost stakeholder participation and support by providing assistance in resource allocation, establishing directives and policies, and strategic decision-making. Continuous communication is crucial in decreasing employees' uncertainty and resistance while increasing acceptance. This includes providing information on the implementation's purpose, process, and timeline, emphasizing the benefits of the cobot for each stakeholder throughout the project stages.

### 1.8. Guidelines for Safe and Effective Human–Robot Collaboration

The literature corpus mentioned above led scholars to develop safe and smooth human–robot collaboration guidelines. Safe and effective human–robot collaboration refers to the successful integration of human and robotic agents within a shared environment or task, designed to optimize performance and efficiency while minimizing risks and hazards. The goal is to establish a harmonious partnership that capitalizes on the strengths of both entities, with an emphasis on worker's health and safety, reliability, and productivity.

To ensure the long-term success of these technological implementations, organizations need to prioritize the safety and psychological well-being of their employees, as well as the efficient and responsible use of resources. Developing clear guidelines for human–robot collaboration promotes a safe and harmonious working environment, fostering employee satisfaction, engagement, and productivity.

Gualtieri et al. [6] developed and evaluated guidelines to facilitate the seamless integration of cobots in organizations. A section of the guidelines is specifically related to organizational measures. Eight guidelines suggest prioritizing the robotic system's safety and reliability to users, framing the robot as a helpful companion, providing training and empowerment, and promoting a sense of responsibility and meaning in the user's work. Furthermore, the guidelines suggest encouraging clear communication from management about the technology's intent and impact, implementing measures to prevent deskilling, and maintaining worker agency and control over delegated tasks [53].

Our study aims to evaluate the perceived impact, implementation solutions, and assessment methods or KPIs for the eight organizational guidelines designed to optimize human–robot interactions. These guidelines focus on safety, training, communication, worker agency, and stakeholder involvement. The study gathers experts' opinions from various fields, such as robotics design, human factors, and organizational psychology, by conducting an online survey. The survey assesses the guidelines' potential impact on workers' psychological well-being, safety, and performance and proposes implementation solutions and KPIs for evaluating their effectiveness.

To our knowledge, no previous studies aimed to provide strategies for companies to ensure employees' psychological well-being, optimize performance, and promote smooth integration of new technologies. The overarching objective of the present contribution is to address factors companies can leverage to address the challenges of robotic technology integration better and to promote a more sustainable and human-centered approach to robotic systems in the workplace. The study uses an online survey to gather expert opinions on the potential impact of the guidelines, proposed implementation solutions, and key performance indicators for evaluating their effectiveness.

## 2. Materials and Methods
### 2.1. Measures

A survey has been developed with the aim of collecting experts' opinions about the organizational guidelines in terms of perceived impact, implementation solutions, and assessment methods or KPIs.

The eight organizational guidelines are the following:

G1. Demonstrate to the user the effectiveness and reliability of safety measures of the robotic system prior to start the interaction.

G2. Make the robotic system perceived by the user as a useful, effective, and reliable companion instead of a competitive entity.

G3. Provide training and empowerment to the user when designing, implementing, and working (e.g., understand the abilities and the process complexity).

G4. Provide measures for experiencing meaning, feeling responsible for outcomes, and understanding the results of the efforts.

G5. Support the management to clearly communicate the changes related to the new technology introduction and its intent, rationale, goals, effects, and commitment.

G6. Implement measures to counteract deskilling of operators when possible and appropriate.

G7. Prevent workers' limited agency, perceived control, and responsibility over the work that the delegation of decisions and tasks to the robotic system may introduce.

G8. Consult users and stakeholders during the hazard identification, risk assessment, and safety measures validation.

*Impact on Workers' Psychological Well-being, Safety and Performance.* Experts were asked to respond the following question: "To what extent do you believe the implementation of this specific Guideline will have an impact on the following aspects?", being "Safety", "Worker's Psychological Well-being", and "Performance". The response format consists of a 5-point rating scale, ranging from "not at all = 1" to "extremely = 5".

*Implementation Solutions.* Experts were asked to briefly describe 1 to 3 possible practical solutions for implementing the guideline according to their knowledge and expertise.

*Assessment Methods and Key Performance Indicators.* Experts were asked to provide assessment methods or Key Performance Indicators (KPIs) that could be used to evaluate each guideline's effectiveness.

### 2.2. Procedure

The data were collected through an online survey. The researchers established an email contact list including experts with different areas of expertise, such as robot design and control, human factors, or work and organizational psychology. The experts are of different nationalities and were identified through the screening of articles' authors who conduct research in the field and through the researchers' professional networks. Additionally, relevant groups and communities (e.g., "Robotics Guru" and "Industrial Robotics") of social networks such as LinkedIn were identified. Thus, invitations to participate in the survey were sent by email and posted on social networks. Experts in different countries completed the survey from February to April 2023. At the beginning of the survey, participants had to confirm that they had read and agreed to the privacy and participation information before completing the survey.

This study is part of a European project funded by the H2020 program. The project, SESTOSENSO, aims to design and validate protocols for a safe, effective, and smooth collaboration with robots. The project has received ethical approval from various academic institutions involved. This study was explicitly approved by the Bioethics Committee of the Alma Mater Studiorum-University of Bologna, following ethical requirements (Prot. n. 0185076) and in compliance with the Declaration of Helsinki.

### 2.3. Participants

The study sample comprises 108 subject matter experts from various countries, representing national and international perspectives. After cleaning the dataset for incomplete and inconsistent responses, a sample of 56 respondents was available. The gender distribution among participants includes 37.5% females, 58.9% males, and 3.6% identifying as non-binary or preferring not to disclose their gender. The average age of the experts is 38.9 years, with a standard deviation of 13.0 years. Participants were asked to state which area of expertise they would indicate as theirs, choosing one or more areas. Table 1 shows the distribution of the participants across different expertise areas and sectors. In addition,

a significant majority of participants (93%) perceive themselves as moderately or highly knowledgeable in the field of robotics.

**Table 1.** Areas of expertise of participants.

| Area of Expertise | *n* | % |
|---|---|---|
| Safety of Machinery | 10 | 17.9 |
| System Integration/Work cell design | 10 | 17.9 |
| Sensor Technology | 11 | 19.6 |
| Software and System Architecture | 11 | 19.6 |
| Simulation and Digital Modeling | 14 | 25.0 |
| Robot Design and Control | 22 | 39.3 |
| Human and Organizational Factors | 27 | 48.2 |
| Human–machine Interface and User Experience | 30 | 53.6 |

Note: The % is calculated on the total sample ($N = 56$) since participants could select more than one answer.

### 2.4. Statistical Analysis

Respondents' opinions on the different impacts of the guidelines were examined using analysis of variance (ANOVA) with Welch's correction and Games-Howell pairwise comparisons for psychological well-being, safety, and performance.

## 3. Results

### 3.1. Perceived Impact on Psychological Well-Being, Safety, and Performance

Table 2 shows the means and standard deviations related to the perceived impact of the guidelines on workers' psychological well-being, safety, and performance. The data highlight the experts' opinions about the potential impact of various organizational guidelines (G1–G8) on the three key aspects. Considering the range of responses, average values above 4 could be deemed as "high impact"; values between 3.9 and 3 could be seen as "moderate impact", and values below 2.9 could be seen as "low impact". G1 (demonstrating safety measures) has a particularly high impact on safety (4.70) and psychological well-being (4.10). This highlights that ensuring users know the effectiveness and reliability of safety measures before interacting with the robotic system might create a strong sense of safety and trust. G2 (perceiving the robot as a companion) has a low impact on safety (2.73) but a high impact on psychological well-being (3.73) and performance (3.82). This suggests that users who see the robotic system as a helpful partner rather than a competitor are likely to feel more comfortable and perform better. G3 (user training and empowerment) has a particularly low impact on performance (2.71). While education and understanding are important, additional factors might be necessary to improve performance. G4 (providing meaning and responsibility) has a high impact on psychological well-being (4.17) and performance (4.17). This shows that fostering a sense of purpose and accountability can significantly improve user satisfaction and productivity. G5 (clear communication) has a moderate impact on psychological well-being (3.57) and performance (3.57). Transparent communication from management about technology changes is important for employee morale and efficiency. G6 (counteracting deskilling) has a moderate impact on safety (3.38), psychological well-being (3.75), and performance (3.13). This suggests that addressing the potential loss of skills among operators is essential for maintaining a positive work environment and an effective workforce. G7 (preventing limited agency) has a moderate impact on safety (3.17), psychological well-being (3.33), and performance (3.17). It is important to ensure that workers maintain a sense of control and responsibility even when tasks are delegated to robots. G8 (consulting users and stakeholders) has a high impact on safety (4.50) and psychological well-being (4.17). Involving users and stakeholders in decision-making can lead to safer, more satisfying work experiences and moderately improved performance.

**Table 2.** Perceived impact on workers' safety, psychological well-being, and performance.

| Organizational Guidelines | Impact on Safety | | Impact on Psychological Well-Being | | Impact on Performance | |
|---|---|---|---|---|---|---|
| | M | SD | M | SD | M | SD |
| G1. Demonstrate to the user the effectiveness and reliability of safety measures of the robotic system prior to start the interaction | 4.70 | 0.48 | 4.10 | 0.74 | 3.80 | 0.42 |
| G2. Make the robotic system perceived by the user as a useful, effective, and reliable companion instead of a competitive entity | 2.73 | 1.19 | 3.73 | 1.10 | 3.82 | 0.87 |
| G3. Provide training and empowerment to the user when designing, implementing, and working (e.g., understand the abilities and the process complexity) | 3.29 | 1.60 | 3.00 | 1.41 | 2.71 | 1.38 |
| G4. Provide measures for experiencing meaning, feeling responsible for outcomes, and understanding the results of the efforts. | 3.17 | 1.17 | 4.17 | 1.17 | 4.17 | 1.17 |
| G5. Support the management to clearly communicate the changes related to the new technology introduction and its intent, rationale, goals, effects, and commitment. | 2.71 | 1.25 | 3.57 | 1.51 | 3.57 | 1.27 |
| G6. Implement measures to counteract deskilling of operators when possible and appropriate. | 3.38 | 0.92 | 3.75 | 0.71 | 3.13 | 1.13 |
| G7. Prevent workers' limited agency, perceived control, and responsibility over the work that the delegation of decisions and tasks to the robotic system may introduce. | 3.17 | 1.03 | 3.33 | 0.89 | 3.17 | 0.84 |
| G8. Consult users and stakeholders during the hazard identification, risk assessment, and safety measures validation. | 4.50 | 0.84 | 4.17 | 0.75 | 3.33 | 1.37 |

We then sought to understand whether there were significant differences in the experts' evaluation of each guideline regarding the perceived impact on workers' psychological well-being, safety, and performance. A one-way ANOVA revealed that there were no significant differences between each guideline in terms of workers' psychological well-being, $F_{(7, 59)} = 1.226$, $p = 0.30$, and performance, $F_{(7, 59)} = 1.546$, $p = 0.17$. For what regards the impact on safety, significant differences were observed, $F_{(7, 59)} = 4.083$, $p = 0.001$. Specifically, multiple comparisons revealed that experts evaluated G1 as having a significantly higher impact on safety compared to G2 ($p = 0.002$), G5 ($p = 0.002$), and G7 ($p = 0.03$). Furthermore, G2 was evaluated as having a significantly lower impact on safety compared to G8 ($p = 0.04$).

### 3.2. Solutions and KPIs for the Organizational Guidelines

This section provides a comprehensive overview of the proposed solutions and their corresponding key performance indicators (KPIs) delineated by subject matter experts for each guideline.

G1. The solutions for implementing the guideline include integrating digital twin technologies, providing demonstration videos, designing supporting documentation, explaining safety measures, implementing a multi-agent supervisory system, offering practical courses, and providing hands-on training. Other solutions include performing work

cycles not for production, having visual indicators for hazards, and providing documentation. Assessment methods or KPIs include subjectively perceived safety, accidents and near misses, risk assessment, quizzes or tests, demonstrations, observations, safety functional, sensor-based monitoring, small-scale study, user satisfaction questionnaire, and user performance.

G2. The solutions for implementing the guideline include designing a dynamic robotic system that adapts to the user, planning robot trajectories with smooth motions, holding workshops with workers, measuring stress levels and perceived safety of the human operator, and using established usability measures. The assessment methods or KPIs for measuring the effectiveness of the implementation include cycle time, general worker performance, user satisfaction, forms for gathering feedback, sensor-based monitoring, and measuring stress level and perceived safety of the human operator.

G3. The solutions for implementing the guideline include conducting a co-creation process with the target user group, using self-assessment simulation tools, providing training courses, implementing the Eyes Principle in panels and task managers, involving a multidisciplinary work group, and conducting external reviews. The assessment methods or KPIs include using questionnaires, measuring stress levels, encouraging user dialogue, and measuring worker satisfaction. According to the experts, it is crucial to ensure that the practical solutions are actionable and have measurable outcomes, and the assessment methods and KPIs should be specific to the guideline and focus on factors such as worker satisfaction, stress level, and usability.

G4. The solutions for implementing the guideline include gathering feedback from workers, providing instantaneous and simple feedback, configuring process plans, defining categories of measures, allowing anonymous suggestions, and conducting post-event analysis. The assessment methods or KPIs include measuring the implementation of the solutions and conducting surveys and questionnaires to gather feedback and assess user satisfaction.

G5. The solutions for implementing the guideline include creating a demonstration video, providing a template and guide, determining the best communication channels, and providing a financial and temporal budget. The suggested assessment methods include checking if the actions have been taken, measuring perceived usefulness, and conducting surveys or interviews to determine worker understanding and agreement with the changes. It is important to ensure that the guidelines address safety, psychological well-being, and performance, and using a rubric could be helpful. Eliciting and factoring worker opinions can improve success in implementing new technologies.

G6. The solutions for implementing the guideline include collecting individual workers' requirements, providing training and refresher courses, implementing a learning mechanism, observing user concentration and attention, assessing user skills, measuring quality and speed of work, and considering organizational limitations. Assessment methods and KPIs include completion time, decrease in accidents, increase in non-faulty products, simulations and surveys for users, statistical properties of worker performance, and regular test runs. Leaving important tasks to people and using smart glasses to display instructions is important. Trials should be used to improve performance.

G7. The solutions for implementing the guideline include establishing collaboration goals, avoiding static robotic behavior, allowing users to choose pre-determined robotic actions, and addressing machine ethics concerns. Assessment methods and KPIs include collaboration level, the incidence of user frustration, number of human–robot interactions, errors and workload of operators, and evaluating the user experience using subjective and objective measures such as the "sense of control" questionnaire. Using cognitive task analysis, expert decision systems, and semi-autonomous mechanisms is also recommended. The overall goal is to ensure smooth transition behavior and effective collaboration between humans and cobots.

G8. The solutions for implementing the guideline include conducting inspections and seminars, encouraging teamwork, using natural language, establishing effective com-

munication channels, demonstrating and discussing machine operation with users, and conducting audits and structured interviews. The assessment methods or KPIs include audit reports, questionnaires to assess user/stakeholder input, effectiveness in reducing errors, number of participants and meetings, participant assessment of risk reduction measures, and SWOT surveys.

## 4. Discussion

The results presented indicate the importance of effectively demonstrating the safety measures of the robotic system prior to user interaction. The quantitative data demonstrates that a combination of transparent communication, fostering a sense of meaning and responsibility, and involving users and stakeholders in decision-making can improve safety, psychological well-being, and performance. While different guidelines may have varying degrees of impact, together they can create a more effective and harmonious work environment involving robotic systems.

The study results showed no significant differences between the guidelines regarding workers' psychological well-being and performance. However, for what regards the impact on safety, significant differences were observed. Experts rated G1 as having a higher impact on safety compared to G2, G5, and G7 and rated G2 as having a lower impact on safety than G8. These results suggest that guidelines that focus on demonstrating the effectiveness and reliability of pre-interaction safety measures, consulting users and stakeholders during risk assessment and validation of safety measures, and reducing the workers' limited agency and perceived control over what is attached to preventing robotic system-delegated work may have a greater impact on preventing safety-related events compared to guidelines aimed at perceiving the robotic system as a useful and effective companion and helping management communicate changes related to technology adoption. These results underscore the importance of considering safety measures from the outset of technology integration and actively involving workers and stakeholders in the process to promote a safe and effective work environment.

The findings suggest several practical solutions for creating a safe and collaborative environment in human–robot interaction, such as digital twin technologies, demonstration videos, and hands-on training. The expected effect is to enhance the users' perceived safety and confidence in the system. These solutions specifically include demonstrating the effectiveness and reliability of safety measures, designing dynamic robotic systems that adapt to users, providing training and empowerment to users, supporting management in communicating changes related to new technology, implementing measures to counteract deskilling of operators, preventing workers' limited agency and perceived control, and consulting users and stakeholders during hazard identification, risk assessment, and safety measure validation. Overall, it is important to ensure that the guidelines address safety, psychological well-being, and performance, and using a rubric could be helpful. The success of these solutions can be assessed through various key performance indicators that can be employed to assess these implementations' success using various methods such as measuring worker satisfaction, stress level, usability, and perceived safety, conducting surveys and questionnaires, and gathering feedback from workers. Ensuring that users feel safe and well-prepared when interacting with the robotic system is crucial to fostering trust and promoting successful human–robot collaboration [54–56].

Another critical aspect is perceiving the robotic system as a helpful companion rather than a competitive entity. To achieve this, implementing dynamic and adaptive robotic systems, smooth trajectory planning, and usability measures can enhance user satisfaction and reduce stress levels. Assessment methods such as cycle time, general worker performance, and sensor-based monitoring provide valuable insights into the effectiveness of these solutions. By focusing on the user's experience and fostering a sense of collaboration, the robotic system's integration into the workplace will likely be more successful and positively received [57–60].

Providing training and empowerment to users is critical in ensuring they can effectively understand and work with the robotic system. Solutions like co-creation processes, self-assessment simulation tools, and multidisciplinary work groups can lead to more informed and satisfied users. Relevant KPIs for measuring the effectiveness of these implementations include questionnaires, stress levels, and worker satisfaction. By prioritizing user engagement and empowerment, implementing new technologies can lead to positive outcomes for both the workers and the organization as a whole [61,62].

Despite the valuable insights provided by the study, certain limitations should be acknowledged. Firstly, the study's scope may not have covered all possible solutions and assessment methods for each guideline. This leaves room for potential gaps in understanding and evaluating the effectiveness of different implementations. Secondly, the study relies heavily on expert opinions and experiences, which may introduce biases or may not be entirely representative of the diverse range of industries and work environments where robotic systems are employed. Additionally, the study's findings may not be generalizable to all types of robotic systems and applications, as different contexts may require unique approaches to safety measures and user interaction. Further research is necessary to address these limitations and expand the applicability of the study's findings across different domains and robotic systems.

Despite its limitations, the study had the valuable merit of highlighting key aspects of human–robot interaction and providing practical solutions for fostering a safe, collaborative, and user-friendly environment. By addressing essential guidelines and offering various implementation strategies and assessment methods, the study serves as a solid starting point for organizations looking to integrate collaborative robotic systems into their operations. The insights gained from this research can guide the development of more effective and inclusive technologies, promoting a harmonious partnership between humans and cobots in the workplace.

Several potential avenues for future research could build upon the findings of this study. Firstly, more research could be conducted to explore the effect of the presented guidelines and suggested practical implementation methods on influencing the relationship between technology integration, well-being, safety, and performance in industrial and other contexts. This could involve conducting experimental studies in real or quasi-real contexts, possibly with longitudinal designs. Longitudinal studies could track changes in these outcomes over time and examine the long-term effects of technology integration on well-being and safety. Furthermore, future studies could investigate how different types of technology or work environments may affect these outcomes. Finally, further research could be done to develop interventions or policies that can help to promote well-being and safety in technology-driven work environments. This could involve testing the effectiveness of different types of training, workplace policies, or other interventions aimed at improving well-being and safety outcomes for workers.

While further research is needed to refine and expand upon these findings, the study's contributions to the field are undeniably significant and have the potential to shape future human–robot collaborations across diverse industries.

## 5. Conclusions

This article contributes to a better understanding of implementing and assessing collaborative robotic systems in organizations, focusing on fostering safe and effective human–robot collaborations. This means that the collaboration between humans and cobots should be designed and executed to minimize potential harm or risks to workers while also optimizing the productivity and performance of the overall system. On the whole, safe and effective human–robot collaboration involves striking a balance between ensuring workers' safety and psychological well-being and achieving the desired productivity and performance outcomes. The implications for sustainability in organizations are multifaceted, as the study highlights key strategies for ensuring workers' psycholog-

ical well-being, optimizing performance, and promoting a smooth integration of new technologies [63].

The study underscores the importance of safety measures and adequate training in promoting long-term retention and job satisfaction among workers, resulting in reduced turnover rates and associated costs. Additionally, perceiving the robotic system as a collaborative entity fosters a more collaborative work environment, ultimately increasing overall productivity and boosting worker morale. Moreover, by prioritizing user empowerment, engagement, and effective communication of changes related to new technology implementations, organizations can better adapt to technological advancements while upholding their commitment to employee well-being [64,65]. By addressing potential issues such as deskilling and worker agency, our study promotes a more sustainable approach to technology integration, where human expertise and creativity are valued alongside automation. Overall, our study emphasizes the importance of sustainable organizations that prioritize the needs of both workers and the environment.

**Author Contributions:** Conceptualization, L.P. and H.B.; methodology, L.P.; investigation, F.F.; resources, L.P.; data curation, F.F.; writing—original draft preparation, F.F.; writing—review and editing, L.P., H.B. and F.F.; supervision, L.P.; project administration, L.P.; funding acquisition, L.P. All authors have read and agreed to the published version of the manuscript.

**Funding:** This paper is granted from the European Commission's HORIZON EUROPE Research and Innovation Actions under the project Sestosenso (grant agreement No 101070310). The material presented and views expressed here are the responsibility of the authors only. The E.U. Commission takes no responsibility for any use made of the information set out.

**Institutional Review Board Statement:** The study received ethical approval by the Bioethics Committee of the Alma Mater Studiorum—University of Bologna (Prot. n. 0185076) and complied with the Declaration of Helsinki (World Medical Association, 2013).

**Informed Consent Statement:** Informed consent was obtained from all subjects involved in the study, outlining participation procedures, study content, data collection purposes, future data dissemination methods, participant rights, and contact information. Participation was voluntary, and participants had the option to withdraw at any time without repercussions. Data collected would be anonymized, and only aggregated data would be used in the analysis.

**Data Availability Statement:** The dataset is available from the corresponding author upon reasonable request.

**Conflicts of Interest:** The authors declare no conflict of interest.

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
