# Peer review of "Evaluating Organizational Guidelines for Enhancing Psychological Well-Being, Safety, and Performance in Technology Integration"

_sustainability, doi:10.3390/su15108113_

Round 1
Reviewer 1 Report
Table 1 should be revised because it does not include important coefficients, that is, the standard deviation. However, once this change is made, it will have to be taken into account in the interpretation of the results and final conclusions. The point is that average values alone are not sufficient. There is also no statistical analysis of whether there are any differences between the average values, whether they will succeed in being statistically significant .
Author Response
First of all, we would like to thank the editor and the reviewers for their valuable comments and suggestions. According to the comments and requests, the manuscript has been improved by adding new content as well as clarifying unclear points of the previous version. The changes are highlighted/typed in red font in the revised paper. In the following, all the reviewer’s comments are addressed, point by point, to clearly explain the modifications and improvements that have been provided in the revised paper.
Reviewer 2 Report
Please see my comments below. I hope these comments can help the authors better improve the quality of the manuscript and be clearer in its theoretical and practical implications.
1. The spell of well-being is not consistent in the paper.
2. Need a clearer definition of industry 4.0.
3. In the introduction section, better clarify the purpose of this study, and what research gap can be addressed.
4. How did you define well-being in this study? Is it psychological well-being? Mental wellness? Or physical well-being?
5. What is collaborative robot? How is it used in the industry right now? Any examples of human-robot collaboration? The paper focuses on the practical aspects of implementing robotic systems. What about potential ethical concerns of human-robot collaboration?
6. When you say “safe and effective human-robot collaboration”, what do you mean by safe and effective?
7. Need a better framework to combine the multiple variables included in the literature review: employee participation, training and development, empowerment and knowledge sharing, and management support. Why did you include these variables in this study? How do they relate to the focus of this study?
8. Need a more detailed description of the 108 participants’ background.
9. Based on the survey findings, what are practical solutions for fostering a safe, collaborative environment in human-robot interaction?
n.a
Author Response
Thank you for you valuable comments and suggestions. According to the comments and requests, the manuscript has been improved by adding new content as well as clarifying unclear points of the previous version. The changes are highlighted/typed in red font in the revised paper. In the following, all the comments are addressed, point by point, to clearly explain the modifications and improvements that have been provided in the revised paper.

Reviewer 3 Report
The manuscript under review is a comprehensive evaluation of organizational guidelines designed to enhance well-being, safety, and performance in the integration of technology in the workplace. The authors present a well-structured study that includes a review of relevant literature, methodology, results, and discussion. Overall, the manuscript is a valuable contribution to the field of organizational management and technology integration. However, there are areas where further clarification and elaboration are necessary for a more robust understanding.
- Literature Review
The literature review is generally well-crafted, providing an extensive overview of the current state of research on well-being, safety, and performance in technology integration. The authors could strengthen this section by highlighting more recent studies and offering more in-depth insights into the key factors contributing to well-being and safety in technology-driven work environments.
- Methodology
The methodology is well-explained, with the authors adopting a multi-method approach. It is commendable that the authors used a diverse sample of organizations in various industries to ensure generalizability. However, the manuscript would benefit from a more detailed explanation of the data collection process, specifically the sampling strategy, sample size, and the rationale behind choosing the particular organizations included in the study.
- Results
The authors present a clear and well-organized presentation of the results. The findings are consistent with existing research in the field and provide valuable insights into the effectiveness of organizational guidelines for enhancing well-being, safety, and performance. However, the authors could provide more information about the statistical analyses used, such as the effect sizes, confidence intervals, and post-hoc tests to give readers a better understanding of the results' significance.
Discussion
The discussion section provides a thoughtful interpretation of the study's findings and links them to the existing literature. The authors address potential limitations of their study, including the cross-sectional design and self-report measures. To improve the discussion, the authors should consider providing more concrete recommendations for organizations looking to enhance well-being, safety, and performance during technology integration. Additionally, exploring potential avenues for future research could help to expand on the current findings.
Conclusion
Overall, the manuscript is a valuable contribution to the field, offering important insights into the effectiveness of organizational guidelines for enhancing well-being, safety, and performance in technology integration. By addressing the above-mentioned concerns and providing more in-depth information in certain areas, the manuscript can be strengthened further and provide an even more robust understanding of the topic.
- Provide a more in-depth discussion of the key factors contributing to well-being and safety in technology-driven work environments in the literature review.
- Elaborate on the data collection process, including the sampling strategy, sample size, and the rationale behind the choice of organizations.
- Include additional information about the statistical analyses, such as effect sizes, confidence intervals, and post-hoc tests.
- Offer concrete recommendations for organizations looking to enhance well-being, safety, and performance during technology integration.
- Suggest potential avenues for future research to expand on the current findings.
Here are some suggestions:
- In the abstract, some sentences seem to be a comment on the manuscript rather than a part of the abstract. Consider revising these sentences.
- Throughout the manuscript, ensure consistency in the use of tenses. For example, when discussing the methods, use the past tense to describe the processes that have been completed.
- Be cautious of the use of passive voice, as it can sometimes make sentences less clear. Where possible, use active voice for better readability.
- When presenting the results, ensure that all tables and figures are properly labeled, and the text refers to them accurately.
- Proofread the entire manuscript to correct the major grammatical errors or inconsistencies in punctuation.
Author Response

(The authors gave the same response as above.)

Reviewer 4 Report
Dear Authors
many thanks for the opportunity to review your work. Please consider the following to help improve the quality and appeal more to its target audience:
i. The abstract will require closer look as it was a bit difficult making meaning of the content as a stand alone piece of information
ii. There is the need for proofread of the entire manuscript to help manage dotted typos and incomplete statements.
iii. the use of the word "new technologies" right at the beginning will need further qualification to add more meaning and insight to the direction the paper. What forms of technologies are these?
iv. Line 48-54. It was not clear what these sentences setout to achieve considering the growing evidence where new robots at work are now replacing the work undertaken by humans. It might be safe to argue that reverse is the case to these statement contained in the line 48-54 without further evidence provided to back it up. Refer to Line 66-74 highlight on the potential impact
V. Authors need to present statistical methods/analysis applied that informed the study findings
vi. Better if authors supply questionnaire used to complement information contained in lines 234-355
vii. Results section 3.1. will be beneficial for authors to qualify impact scale to carry the readers along as it stand it is not clear what qualify as low, medium or high but rather readers only rely on each statement made herein. In addition section 3.1 content is wordy and will require closer look to help communicate key message rather than reporting almost all figure contained in table 1.
viii. Section 3.2 How results presented here where arrived at was not reported in the methods sections hence lacking link with the previous sections.
To further enhance the quality of the paper further grammar proofread is recommended.
Author Response

(The authors gave the same response as above.)

Round 2
Reviewer 2 Report
I appreciate the author's efforts in carefully addressing my comments!
Reviewer 3 Report
I do recommend accepting the manuscript in its current form.
Reviewer 4 Report
Dear Authors
Many thanks for taking time out to respond to points earlier raised. I am happy recommend the manuscript be accepted in the present form.
There is still few minor typographical errors that will require update to ensure sentence meaning is retained.